# Dupilumab as Therapeutic Option in Polysensitized Atopic Dermatitis Patients Suffering from Food Allergy

**DOI:** 10.3390/nu16162797

**Published:** 2024-08-22

**Authors:** Alvise Sernicola, Emanuele Amore, Giuseppe Rizzuto, Alessandra Rallo, Maria Elisabetta Greco, Chiara Battilotti, Francesca Svara, Giulia Azzella, Steven Paul Nisticò, Annunziata Dattola, Camilla Chello, Giovanni Pellacani, Teresa Grieco

**Affiliations:** 1Dermatology Unit, Department of Clinical Internal Anesthesiological and Cardiovascular Sciences, “Sapienza” University of Rome, 00161 Rome, Italy; alvise.sernicola@unipd.it (A.S.); emanuele.amore@uniroma1.it (E.A.); rizzutogiuseppe98@gmail.com (G.R.); alessandrarallo93@gmail.com (A.R.); mariaelisabetta.greco@uniroma1.it (M.E.G.); chiara.battilotti@gmail.com (C.B.); francescasvara@gmail.com (F.S.); azzella.1701118@studenti.uniroma1.it (G.A.); steven.nistico@uniroma1.it (S.P.N.); nancydattola@gmail.com (A.D.); pellacani.giovanni@uniroma1.it (G.P.); teresa.grieco@uniroma1.it (T.G.); 2Dermatology Unit, Department of Medicine (DIMED), University of Padua, 35121 Padova, Italy

**Keywords:** dupilumab, food allergy, comorbidity, immunotherapy

## Abstract

IgE-mediated food allergy is characterized immunologically by a type 1 immune response triggered upon exposure to specific foods and clinically by a broad range of manifestations and variable severity. Our understanding of food allergy within the allergic march of atopic dermatitis (AD) is still incomplete despite the related risk of unpredictable and potentially severe associated reactions such as anaphylactic shock. The aim of this pilot study was to investigate the effects of dupilumab, an IL-4/IL-13 monoclonal antibody approved for AD, on the allergic sensitization profile of patients with AD and type 1 hypersensitivity-related comorbidities, including oral allergy syndrome, anaphylaxis, and gastrointestinal disorders. We conducted an observational pilot study with a longitudinal prospective design, enrolling 20 patients eligible for treatment with dupilumab. Laboratory exams for total serum IgE, specific IgE, and molecular allergen components were performed at baseline and after 16 weeks of therapy. Our results demonstrate a statistically significant decrease in molecular components, specific IgE for trophoallergens, and specific IgE for aeroallergens following treatment with dupilumab. We suggest that modulating type 2 immunity may decrease IgE-mediated responses assessed with laboratory exams and therefore could minimize allergic symptoms in polysensitized patients. Upcoming results of randomized controlled trials investigating dupilumab in food allergy are highly anticipated to confirm its potential effect in the treatment of IgE-mediated food allergies.

## 1. Introduction

Atopic dermatitis (AD) is an eczematous skin disease characterized by chronic and relapsing episodes associated with pruritus and responsible for a severe impairment in quality of life [1]. For moderate to severe AD, current treatment recommendations include biologic drugs and innovative small molecules, as outlined in the 2022 European Academy of Dermatology and Venereology (EADV) guidelines [2].

Atopic dermatitis (AD) is identified as the primary and most frequent manifestation of the atopic march, occurring in 15–20% of children [3]. This condition may persist into adolescence, adulthood, and old age, and commonly coexists with various comorbidities including asthma, rhino-conjunctivitis, eosinophilic esophagitis, and food allergies [4,5]. Within the context of type 1 immune responses, food allergies emerge as a critical life-threatening issue for patients and their management constitutes a challenge for physicians. Clinical outcomes range from oral allergy syndrome (OAS) to severe and generalized reactions such as anaphylactic shock, which may unexpectedly develop at any stage of life in around 5% of those affected [6]. In pediatric subjects, peanuts, milk, and eggs allergies can be more often concerning; in adults, peanuts, seafood and fish, and certain fruits (peach and apple) are the main allergens involved in food allergies, but do not directly predict a severe course of AD [7]. Moreover, OAS is based on the sensitization to respiratory allergens that display cross-reactivity with specific foods. Food allergy represents a substantial health burden for patients, determining several restrictions in daily activities and behaviors as well as a global increase in health-related costs. Eventually, this condition causes a considerable impairment in the quality of life of affected individuals. The most recent approach to the management of food allergies uses non-allergen-specific immunotherapy with biologics that reduce type 2 inflammation. The US Food and Drug Administration (FDA) approved Oral Immunotherapy (OIT) for peanut allergies and, in 2024, the anti-IgE monoclonal antibody omalizumab, reflecting progress in allergy treatments [8,9]. Dupilumab, an anti-interleukin-4/13 receptor monoclonal antibody that inhibits the differentiation of naïve T-cells into the Th2 subtype, is presently endorsed for the management of moderate to severe atopic dermatitis (AD) [10]. Research on this biologic monoclonal antibody for treating type 1 food allergy manifestations in phase 2 and phase 3 clinical trials is under way.

This study focuses on a relatively underexplored area: the impact of dupilumab therapy on the food allergy sensitization profiles among polysensitized AD subjects. Given the complexity of AD and its association with various types of sensitizations, understanding how dupilumab therapy influences these profiles is crucial for tailoring individualized treatment approaches and improving patient outcomes.

## 2. Materials and Methods

We conducted a prospective observational pilot study aimed at evaluating the longitudinal changes in food allergy sensitization profiles in a cohort of 20 polysensitized patients with AD who have been receiving dupilumab therapy over a span of one year without interrupting or discontinuing the treatment. Concurrent administration of topical steroids and calcineurin inhibitors was allowed during dupilumab therapy for AD.

### 2.1. Study Population

This study included adult patients who were referred to our dermatology unit with a diagnosis of moderate to severe AD and who received dupilumab for at least 16 weeks without interruptions. Out of 85 patients under treatment with dupilumab in our outpatient clinic from January 2019 to December 2021, the study cohort enrolled 20 participants who met the following inclusion criteria:Subjects of both sexes aged 18 years or older (11 females and 9 males).Individuals diagnosed with moderate to severe AD, as determined by clinical assessment. The severity of AD was measured using the eczema area and severity index (EASI), dermatologic life quality index (DLQI), and pruritus numeric rating scale (NRS), since these are the required indices to assess eligibility for prescribing dupilumab.Candidates deemed suitable for dupilumab treatment at the scheduled approved dosage of a 600 mg loading dose followed by 300 mg every 2 weeks.A documented personal history of severe food allergy, characterized by one or more of the following manifestations:○Systemic anaphylaxis, indicating a severe and potentially life-threatening allergic reaction.○OAS, reflecting localized allergic reactions in the mouth and throat shortly after eating certain foods.○Gastrointestinal symptoms indicative of type 1 hypersensitivity, including abdominal pain, vomiting, and diarrhea, following ingestion of specific allergens.Among atopic comorbidities related to type 2 inflammation, allergic rhino-conjunctivitis was reported in 20 patients, a history of asthma was reported in 6 subjects, and none had a diagnosis of eosinophilic esophagitis.

### 2.2. Allergy Assessment Protocol

Following the European Academy of Allergy and Clinical Immunology (EAACI) guidelines [11], all participants in the study were subjected to a comprehensive food allergy protocol. This protocol was designed to accurately assess and monitor the sensitization profiles of each subject to various food allergens. The process involved a series of steps to ensure a thorough evaluation of each patient’s allergic responses and sensitivities:Skin Prick Tests (SPTs) with respiratory and food allergen extracts (ALK-Abelló A/S, Hørsholm, Denmark).Total serum IgE using an ImmunoCAP Total IgE test with the Phadia Laboratory System (Thermo Fisher Scientific Inc., Waltham, MA, USA). Although a control group of healthy participants was not considered for our study, reference values for total IgE are considered up to 100 kU/L [11].Specific IgE for food and respiratory allergens using the ImmunoCAP fluorescence enzyme immunoassay (FEIA) system (Thermo Fisher Scientific Inc., Waltham, MA, USA); food allergens were tested according to anamnesis and included peanut, hazelnut, almond, walnut, cereals, and peach; the major respiratory allergens tested in all patients were *Cupressus sempervirens*, *Dermatophagoides pteronyssinus*, *Dermatophagoides farina*, *Lolium perenne*, *Parietaria Judaica*, and *Poa pratensis*. Results > 0.35 kUA/L are considered positive for allergen sensitization to the specific allergen. Moreover, the sensitivity reported in the literature for ImmunoCAP is between 62 and 88% [12].Concentration of IgE specific for molecular determinants for component-resolved diagnosis using ImmunoCAP FEIA (Thermo Fisher Scientific Inc., Waltham, MA, USA); tested molecules included Pru p 1, Pru p 3, Ara h 1, Ara h 2, Ara h 3, Ara h 9, Cor a 8, Tri a 14, and Tri a 19.

Serum samples were obtained from all patients at the end of a visit before the initiation of treatment (T0) and after 16 weeks of dupilumab (T1), and, considering the individual appointment schedule of each patient that prevented collecting samples from several patients at the same time, were sent on the day of collection to our hospital’s central diagnostic laboratory for routine evaluation as follows. Peripheral venous blood in the non-fasting condition was collected into closed-system BD Vacutainer SST II Advance 5.0 mL tubes (BD, Franklin Lakes, NJ, USA) and allowed a minimum of 30 min to clot. Afterwards, it was centrifuged at 3000 g for 5 min with standard acceleration and soft deceleration settings on Medifuge Centrifuge (Thermo Fisher Scientific Inc., Waltham, MA, USA) at room temperature between 20 and 25 °C, and stored as specified by the manufacturer’s instructions until the assay was performed according to the routine laboratory workflow.

Total and specific IgE were measured using the same method and analytical system for all patients. Results were expressed in kilounits per liter (kU/L) for total IgE and in kilounits of allergen-specific IgE per liter (kUA/L) for specific IgE.

The overall limit of quantitation for total IgE antibodies and for allergen-specific IgE antibodies is 2 kU/L and 0.1 kUA/L, respectively (analytical sensitivity). For both tests, the cross-reactivity with other human immunoglobulins is non-detectable at physiological concentrations of IgA, IgD, IgM, and IgG (analytical specificity) [11,13].

To monitor changes in food allergy sensitization profiles over the course of dupilumab therapy, the allergy assessment protocol was repeated at a defined interval during the 1-year follow-up period (T1 = 16 weeks of dupilumab therapy). These repeat assessments allowed for the evaluation of any shifts in allergen sensitivities following dupilumab treatment.

Pediatric subjects with AD and food allergy were excluded due to the different doses and schedules of administration of dupilumab according to age.

The clinical and demographic characteristics of our patient population are summarized in Table 1.

### 2.3. Statistical Analysis

A descriptive analysis was performed on the data collected. Quantitative variables were expressed as median and range (minimum; maximum) and qualitative variables as frequency and percentage. To compare values of variables between T0 and T1, the authors employed the Wilcoxon signed rank test, a nonparametric test for paired samples. Statistically significant differences were considered for *p* < 0.05. Statistics were performed using Excel version 16 (Microsoft Corporation, Redmond, WA, USA).

All participants provided written informed consent prior to their inclusion in the study, understanding the expected benefits and potential risks of treatment with dupilumab. Ethical review was waived by our local Institutional Review Board for this study due to the observational nature of this study not requiring specific approval.

## 3. Results

In the sample of patients enrolled for this study, laboratory measurements were compared between baseline (T0) and 16 weeks following the initiation of dupilumab. Total serum IgE decreased from a median value of 654 kU/L at T0 to 446 kU/L at T1, showing a statistically significant difference (*p* = 0.049) (Table 2).

In the same timeframe, the median of all specific IgE for respiratory allergens decreased from 6.76 kUA/L to 1.14 kIUA/L (*p* = 0.00001) and the median of specific IgE for food allergens decreased from 8.02 kUA/L to 1.48 kUA/L (*p* = 0.00012). Median values for the major respiratory allergens are also reported in Table 3.

Finally, the assessment of the sum of molecular food allergens for component-resolved diagnosis showed a reduction from a median of 0.2 kUA/L at T0 to a median of 0.1 kUA/L at T1 (*p* = 0.00001). Modifications of values for single molecular allergens are reported in Table 4, highlighting a dramatic reduction during treatment that applies in particular to the values of lipid transfer proteins (Pru p 3, Ara h 9, Cor a 8), which are predictive markers for severe systemic manifestations and anaphylaxis as well as for oral allergy syndrome.

A detailed dietary history was obtained from each patient at baseline to confirm the correlation between allergic manifestations following food intake and the positive values of the related specific IgEs. Although our patients did not undergo the oral food challenge (OFC) while under dupilumab treatment, they were monitored to document any adverse reactions, and no clinical symptoms associated with food exposure were reported throughout the study.

## 4. Discussion

The research for effective therapies in the management of food allergies, including biologics and combination strategies with allergen immunotherapy, is still ongoing and reflects an unmet therapeutic need in this context.

Current Global Allergy and Asthma European Network’s (GA2LEN) recommendations for managing food allergy suggest dietary avoidance of the responsible foods (passive approach) or substituting food consumption and the administration of emergency drugs (reactive approach) in case of accidental exposure [14,15,16].

As far as traditional medications are concerned, the use of oral anti-H1 antihistamines provides symptomatic relief for itching and wheals, while the use of systemic corticosteroids is restricted to patients suffering more severe forms or eosinophilic gastroenteritis. Intramuscular adrenaline is the emergency drug for patients at risk of anaphylaxis [17]. Strategies to obtain an immune tolerance in affected patients are based on allergen-specific immunotherapy (AIT), which consists of standardized allergen extract administration with sequential increasing doses. Oral allergen immunotherapy (OIT), epicutaneous immunotherapy (EIT), or subcutaneous immunotherapy (SIT) represent currently available routes for the management of peanut food allergy, as well as second-class trophoallergen sensitization cross-reacting with aeroallergens like birch. An inherent limitation of the allergen-specific approach is the difficulty in managing polysensitized subjects, with an increase in the number of treatments and the risk of a short duration of desensitization [18].

Non-allergen-specific immunotherapy with biologic drugs against IgE is a novel approach to overcome this limitation, as well as to lower the risk of adverse events during AIT, when used in combination regimens. In 2003, the first study was conducted with TNX-901, a monoclonal anti-IgE antibody, as monotherapy on subjects suffering from peanut allergy. The results of this study demonstrated an increase in the sensitivity threshold to the oral challenge; however, long-term desensitization has never been studied and the drug has not been approved [19].

Subsequently, omalizumab was used experimentally as monotherapy for the treatment of peanut allergy and in clinical protocols as an adjunctive therapy to peanut and milk oral immunotherapy [20,21,22,23,24]. A study by Fiocchi et al. on subjects suffering from severe asthma treated with omalizumab showed the effects of the drug on a subgroup of 15 patients with food allergies. After 4–6 months of treatment, nine patients were able to reintroduce foods into their diet without reactions. Furthermore, reactions due to accidental exposure to food were reduced by 95.7% during treatment [25]. Omalizumab has demonstrated faster and safer oral desensitization without offering a definite advantage in terms of sustained/permanent tolerance. The rapid non-specific desensitization response to trophoallergens observed may be due to the early inhibition of basophils which would play a role in acute allergic reactions [24]. A phase 3 trial with omalizumab and oral immunotherapy for multiple foods (peanuts and at least two other foods including milk, egg, wheat, walnut, cashew, and hazelnut) enrolled 225 subjects (NCT03881696). Anti-IgE treatment can modify not a single but multiple allergens at a time, as the signaling pathways are shared, dependent on basophils in the acute phase and on mast cells in the long term. About one-third of individuals with food allergies are sensitized to multiple foods. Currently, the experimental combination of omalizumab and OIT appears to be the most effective method to reduce sensitization in polysensitized patients [26].

The US FDA approved omalizumab in February 2024 for the treatment of food allergies in children aged 1 year or older and adults. Clinical trials demonstrated that the anti-IgE monoclonal antibody was effective over a 16-week period in reducing allergic reactions to peanuts, cashew, milk, egg, walnut, wheat, and hazelnut compared to a placebo. This treatment resulted in an increased reaction threshold for peanut and other common food allergens. While the initial results are promising, continued research is necessary to assess the long-term outcomes and sustainability of the benefits conferred by omalizumab. Monitoring patients over an extended period will provide insights into the durability of its effects. In summary, the potential of omalizumab for treating multiple food allergies introduces exciting possibilities for improving the lives of individuals, especially young children, living with these challenging conditions. However, ongoing research, careful consideration of safety aspects, and a holistic understanding of the broader implications are crucial to support widespread adoption in clinical practice [26].

Recently, clinical trials have investigated other signaling pathways involved in the allergic response beyond IgE blockade. The expression of IL4 induced by allergens in patients clinically allergic to milk and sensitized with IgE to milk and peanuts suggests a key role of signaling mediated by this cytokine in food allergies [21,22,23,24,27]. Moreover, exposure to food allergens causes, through an IgE-mediated binding to mast cells and basophils, the release of type 2 proinflammatory mediators (IL-4, IL-13, IL-19, and related chemokines) responsible for tissue response and associated symptoms. Thus, the therapeutic inhibition of IL-4/IL-13 signaling with dupilumab is expected to lead to a reduction in the recruitment of intestinal mast cells with consequent attenuation of the clinical expression related to food allergy [27,28].

A 2018 report provided the proof of concept for the possibility to induce food tolerance with dupilumab. The authors described the case of a patient affected by severe AD and food allergy, diagnosed by a positive pattern for lipid transfer proteins and a clinical history of systemic anaphylaxis after the ingestion of corn and walnuts. After three months of treatment with dupilumab, the patient experienced tolerance after two episodes of accidental ingestion of food which had previously caused allergic reactions [27].

A 2019 clinical trial has investigated the trend of total and specific IgE levels (for extracts of *Malassezia*, for *S. aureus* enterotoxin, and for 112 allergens by ImmunoCAP ISAC multiplex assay) in 19 patients with AD treated with dupilumab. The authors demonstrated a statistically significant reduction in serum total IgE along with the total sum of specific IgE measured by ISAC after 16 weeks of treatment. However, a significative correlation between the reduction in total IgE or specific IgE for *S. aureus* and *Malassezia* and the clinical response of AD was not observed [29].

Dupilumab is under clinical investigation for peanut allergy, with two clinical trials already completed in pediatric patients with peanut allergy—one in association with OIT (NCT03682770) and the other in monotherapy (NCT03793608)—and additional trials are currently in progress to assess dupilumab in association with OIT for cow’s milk allergy (NCT04148352) and for multiple food allergens (NCT03679676). Results are available for 24 patients aged 6–17 years with peanut allergy that were randomized to dupilumab 300 mg every 2 weeks or placebo. The study results showed that, after 24 weeks, only 8.3% (IC95, 1.03–27.00%) of patients treated with dupilumab were able to pass a Double-Blind Placebo-Controlled Food Challenge (DBPCFC) with a cumulative dose of peanut protein (444 mg) and 41.7% required rescue use of epinephrine during DBPCFC (NCT03793608). These results suggest that the association of dupilumab with immunotherapy may improve outcomes; however, results from a second study that was completed using dupilumab in association with peanut OIT are still not available (NCT03682770). Furthermore, a phase II clinical trial conducted by Corren et al. in 2021, involving 103 adult patients with grass pollen allergies, examined the efficacy of combining dupilumab with SIT. The study findings suggest that this combination may enhance the tolerability of immunotherapy. However, it did not significantly reduce nasal symptoms following allergen exposure compared to immunotherapy alone [30].

Our study was conducted with a duration of 16 weeks of treatment, which is similar to the trials already present in the literature which evaluated biological drugs in monotherapy for the management of multiple allergies, showing a statistically significant reduction in the total sum of specific IgE for trophoallergens and molecular food allergens for component resolved diagnosis. However, our study does not provide data about variations in the sensitization profile in the long-term or after an eventual interruption of treatment with dupilumab. Considering the mechanism of action of dupilumab, it is possible to hypothesize that the reduction in the variables observed in our study would revert after interruption of treatment.

## 5. Conclusions

AD is rightfully considered a multisystemic disorder of which food allergy constitutes a primary comorbidity that is thought to be underrecognized, leading to the development of concomitant gastrointestinal complications. Therefore, a comprehensive allergology assessment is necessary to achieve a correct diagnosis in subjects with AD. Moreover, a tailored approach in the treatment of AD is feasible only by recognizing all concomitant atopic manifestations and the related therapeutic needs. The results of the present study further support the adoption of a so-called comorbidity-driven approach to the treatment of AD. The recognition of comorbidities related to type 2 immunity including the diagnosis of food allergy, according to our results, can therefore guide therapeutic choices to favor dupilumab.

However, it is presently unknown to what extent and how long allergen tolerance can persist if treatment with dupilumab is eventually interrupted. This limitation is inherent to the current chronic use of the drug, and future studies are necessary to elucidate this aspect. Moreover, the role of differential administration schedules of dupilumab is still investigational, and dose-spacing regimens every 4 weeks are being studied in adolescents with AD [31]. Additional limitations of our study are the small sample size due to the selectivity of the inclusion criteria and the inability to perform an OFC test, which is considered the diagnostic gold standard, even though all patients were on unrestricted diets and no adverse events were registered during the study. Our results have demonstrated a statistically significant reduction in IgE specific for trophoallergens and for their molecular determinants after dupilumab treatment in polysensitized patients. We support that the modulatory effect of dupilumab on type 2 immunity can reduce IgE-mediated responses, as evidenced through laboratory tests in our study, and could consequently determine the attenuation of allergic symptoms. The results of future randomized clinical trials on a broad sample of patients are needed to confirm our preliminary observations. Moreover, dietary restrictions were not prescribed to prevent a concomitant effect on the decrease of specific IgE due to elimination of the implicated food allergens.

## Figures and Tables

**Table 1 nutrients-16-02797-t001:** Clinical and demographic characteristics of the study population.

Variable	Value
Sex [frequency (%)]	
female	11 (55%)
male	9 (45%)
Age [median (min;max)]	27 (18;50) years
Special sites of AD [frequency (%)]	
Head and neck	12 (60%)
Hands	6 (30%)
genitals	2 (10%)
EASI [median (min;max)]	
T0	24 (24;47)
T1 (16 weeks)	3.5 (0;12)
Itch NRS [median (min;max)]	
T0	8 (7;10)
T1 (16 weeks)	1.5 (0;6)
DLQI [median (min;max)]	
T0	13 (7;25)
T1 (16 weeks)	2.5 (0;6)
Manifestation of food allergy	
Systemic anaphylaxis	6 (30%)
Oral allergy syndrome	8 (40%)
Gastrointestinal symptoms	6 (30%)

**Table 2 nutrients-16-02797-t002:** Values of total serum IgE between baseline (T0) and 16 weeks of dupilumab (T1) assessed using ImmunoCAP fluorescence enzyme immunoassay.

	T0 (kU/L) Median (Min; Max)	T1 (kU/L) Median (Min; Max)
**Total IgE**	654 (384; 4740)	446 (118; 2260)

**Table 3 nutrients-16-02797-t003:** Values of IgE for major respiratory allergens between baseline (T0) and 16 weeks of dupilumab (T1) assessed using ImmunoCAP fluorescence enzyme immunoassay.

Allergen (sIgE)	T0 (kUA/L) Median (Min; Max)	T1 (kUA/L) Median (Min; Max)
*Cupressus sempervirens*	5.12 (0.58; 28.8)	2.78 (0.19; 10.9)
*Dermatophagoides pteronyssinus*	24.5 (0.15; 165)	11.55 (0.52; 52.5)
*Dermatophagoides farinae*	35.2 (0.53; 44.5)	10.3 (0.6; 41.2)
*Lolium perenne*	10.22 (1.32; 63.6)	2.245 (0.86; 57.7)
*Parietaria judaica*	1.09 (0.1; 100)	0.31 (0.1; 96.8)
*Poa pratensis*	11.55 (0.99; 100)	3.305 (1.04; 69.3)
Sum of sIgE *	6.76 (0.1; 165)	1.14 (0.1; 96.8)

* This row considers all specific IgEs for respiratory allergens.

**Table 4 nutrients-16-02797-t004:** Values of IgE for molecular food allergens between baseline (T0) and 16 weeks of dupilumab (T1) assessed using ImmunoCAP fluorescence enzyme immunoassay.

Molecular Component (sIgE)	T0 (kUA/L) Median (Min; Max)	T1 (kUA/L) Median (Min; Max)
Pru p 1	0.1 (0.1; 0.12)	0.1 (0.1; 0.1)
Pru p 3	1.05 (0.1; 75.9)	0.465 (0.16; 1.26)
Ara h 1	0.1 (0.1; 0.17)	0.1 (0.1; 0.1)
Ara h 2	0.22 (0.1; 0.42)	0.11 (0.1; 0.18)
Ara h 3	0.1 (0.1; 0.37)	0.1 (0.1; 0.1)
Ara h 9	0.675 (0.1; 37.9)	0.62 (0.1; 7.5)
Cor a 8	0.41 (0.1; 27.6)	0.125 (0.1; 2.24)
Sum of sIgE *	0.2 (0.1; 75.9)	0.1 (0.1; 7.5)

* This row considers all specific IgEs for molecular components.

## Data Availability

The data that support the findings of this study are available from the corresponding author upon reasonable request.

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
