# Peer review of "Dupilumab as Therapeutic Option in Polysensitized Atopic Dermatitis Patients Suffering from Food Allergy"

_nutrients, 2024, doi:10.3390/nu16162797_

Round 1

Reviewer 1 Report

Comments and Suggestions for Authors

Some statements from the abstract and main text  need to be checked and completed . Line 16 - Food allergies can be IgE or non-IgE mediated ; definition has to be completed . Line 19-22 - is too long ,needs to be rephrased ,preferably according to allergic march- respiratory and food allergy . Oral allergy syndrome is based on respiratory allergy and cross-reactivity with some foods , it has to be more clearly defined. Line 46-47 -the 50% risk of anaphylactic shock is too high, has to be checked ! Line 48-49 -please check the responsible foods in children - milk and eggs and in adults - peanuts, seafood, fish , some fruits . Again the oral food syndrome seems to be misunderstood- it does not predict a severe course of AD. Line 93 - specific IgE to foods .Line 148- delete" of the ". Line 165-166 can be deleted. Discussion section can be better structured , focusing on the relevant studies on dupilumab. Limitations are repeated in the Conclusions 

Comments on the Quality of English Language

The quality of English language  is good 

Author Response

Reviewer 1

We thank Reviewer 1 for the valuable feedback on our manuscript. We have made extensive revisions to address all the comments and suggestions raised. Specifically:

Some statements from the abstract and main text  need to be checked and completed . Line 16 - Food allergies can be IgE or non-IgE mediated ; definition has to be completed .

  • Thank you for your comment, we have rewritten the definition in the text. Please see: “Food allergy can be IgE or non-IgE mediated. Food allergy characterized by an IgE-mediated Type 1 immune response is triggered upon exposure to specific foods. Non-IgE mediated food al-lergies, on the other hand, involve different immune mechanisms that do not rely on IgE antibod-ies.”

Line 19-22 - is too long ,needs to be rephrased ,preferably according to allergic march- respiratory and food allergy .

  • Thank you, this sentence has been rephrased according to suggestions: “Our understanding of food allergy within the allergic march of atopic dermatitis (AD) is still in-complete, despite the related risk of unpredictable and potentially severe associated reactions such as anaphylactic shock.”

Oral allergy syndrome is based on respiratory allergy and cross-reactivity with some foods , it has to be more clearly defined.

  • Thank you, we have added a relevant statement in the text: “Moreover, OAS is based on the sensitization to respiratory allergens that display cross-reactivity with specific foods.”

Line 46-47 -the 50% risk of anaphylactic shock is too high, has to be checked !

  • Thank you, the risk was mistakenly reported as 50% instead of 5%: “…anaphylactic shock, which may unexpectedly develop at any stage of life in around 5% of those affected”

Line 48-49 -please check the responsible foods in children - milk and eggs and in adults - peanuts, seafood, fish , some fruits . Again the oral food syndrome seems to be misunderstood- it does not predict a severe course of AD.

  • Thank you for your clarification, we have edited the text accordingly: “…in adults peanuts, seafood and fish, and certain fruits (peach and apple) are the main allergens involved in OAS but do not directly predict a severe course of AD [7]. Moreover, OAS is based on the sensitization to respiratory allergens that display cross-reactivity with specific foods.”

Line 93 - specific IgE to foods .

  • Thank you, we changed to: “Specific IgE to food allergens”

Line 148- delete" of the ".

  • We apologize for this mistake, we have deleted the repetition.

Line 165-166 can be deleted.

  • Thank you, these have been deleted.

Discussion section can be better structured , focusing on the relevant studies on dupilumab.

  • Thank you for your suggestion: we have restructured the discussion section to improve focus on non-allergen-specific treatments: the role of anti-IgE monoclonal antibodies is discussed in depth considering the recent approval of omalizumab in this context and available evidence on dupilumab is reported. Please see the discussion section in the revised manuscript.

Limitations are repeated in the Conclusions

  • We have removed the limitations from the discussion and now they are reported only in the conclusions.

Reviewer 2 Report

Comments and Suggestions for Authors

The authors examined the effect of dupilumab on allergic sensitization profiles of 20 selected subjects. All subjects, as mentioned in the Materials and Methods section, were managed with a controlled administration of food allergens to assess and prevent allergic reactions. The follow-up period was 16 weeks (one year since the dupilumab therapy initiation). They report decreases in total serum IgE, allergen-specific IgE, and respiratory allergen sensitivity over the course of treatment and significant improvement of clinical parameters.

Specific comments:

1.    Pediatric subjects with AD and food allergy were excluded due to the different doses and schedules of administration of dupilumab according to age. The scheduled approved dosage in this study was the 600 mg loading dose followed by 300 mg every 2 weeks for one year treatment period of adults. The paper by Blauvelt et al, Am J Clin Dermatol, 2022, 23:365-383 defines long term efficacy and safety of dupilumab in adolescent subjects (mean age 14.7 years). Subcutaneous dupilumab administration in a dose of 300 mg was used once in 4 weeks for 40 weeks period. Those are lower dosages over the time of treatment, and they worked. Why 2 weeks period in drug administration used in this study?

2.    The existing comorbidities are not clearly defined for the group of patients (for example,

asthma – n=?, rhino-conjunctivitis – n=?, eosinophilic esophagitis – n=?, etc).

3.    It is unclear when the patients in this study were discontinued from dupilumab.

4.    It is unclear if patients used topical steroids (or systemic steroids, or other forms of medication) during dupilumab treatments.

5.    Missing information on some clinical outcome parameters such as Scoring Atopic Dermatitis (SCORAD) and affected body surface area (BSA).

6.    Were median values for the major respiratory allergens reported in Table 2 measured by skin prick test?

Author Response

Reviewer 2

The authors examined the effect of dupilumab on allergic sensitization profiles of 20 selected subjects. All subjects, as mentioned in the Materials and Methods section, were managed with a controlled administration of food allergens to assess and prevent allergic reactions. The follow-up period was 16 weeks (one year since the dupilumab therapy initiation). They report decreases in total serum IgE, allergen-specific IgE, and respiratory allergen sensitivity over the course of treatment and significant improvement of clinical parameters. 

Dear Reviewer 2,

Thank you for your careful review of our manuscript and for the constructive comments.

We have carefully considered each issue raised, please find our responses provided point by point below:

Specific comments:

  1. Pediatric subjects with AD and food allergy were excluded due to the different doses and schedules of administration of dupilumab according to age. The scheduled approved dosage in this study was the 600 mg loading dose followed by 300 mg every 2 weeks for one year treatment period of adults. The paper by Blauvelt et al, Am J Clin Dermatol, 2022, 23:365-383 defines long term efficacy and safety of dupilumab in adolescent subjects (mean age 14.7 years). Subcutaneous dupilumab administration in a dose of 300 mg was used once in 4 weeks for 40 weeks period. Those are lower dosages over the time of treatment, and they worked. Why 2 weeks period in drug administration used in this study? 

- thank you for this comment, dupilumab was chronically administered every two weeks according to the registered dosing in adults with AD. We have added a statement with the relevant reference in the conclusions: “Moreover, the role of differential administration schedules of dupilumab is still inves-tigational and dose-spacing regimens every 4 weeks are being studied in adolescents with AD.”

  1. The existing comorbidities are not clearly defined for the group of patients (for example, 

asthma – n=?, rhino-conjunctivitis – n=?, eosinophilic esophagitis – n=?, etc).

- Thank you for your comment. We have added a statement on the comorbidities to the methods section: “Among atopic comorbidities related to type 2 inflammation, allergic rhi-no-conjunctivitis was reported in 20 patients, a history of asthma was reported in 6 subjects, and none had a diagnosis of eosinophilic esophagitis.”

  1. It is unclear when the patients in this study were discontinued from dupilumab. 

-  Thank you, we have added the following sentence to avoid a potential source of confusion. See in the methods section: “, without interrupting or discontinuing the treatment”

  1. It is unclear if patients used topical steroids (or systemic steroids, or other forms of medication) during dupilumab treatments. 

- Thank you, the following sentence was added to explain this: “Concurrent administration of topical steroids and calcineurin inhibitors was allowed during dupilumab therapy for AD.”

  1. Missing information on some clinical outcome parameters such as Scoring Atopic Dermatitis (SCORAD) and affected body surface area (BSA).

- Thank you, we have explained our choice of clinical parameters in the methods section: “The severity of AD was measured using eczema area and severity index (EASI), dermatologic life quality index (DLQI) and pruritus numeric rating scale (NRS), since these are the required indices to assess eligibility for prescribing dupilumab.”

  1. Were median values for the major respiratory allergens reported in Table 2 measured by skin prick test?

- Thank you, we have clarified in the description of table 2 that these have been measured with immunofluorescence. Please see: “Table 2. Values of IgE to major respiratory allergens between baseline (T0) and 16 weeks of dupilumab (T1) assessed using ImmunoCAP fluorescence enzyme immunoassay.”

Reviewer 3 Report

Comments and Suggestions for Authors

My comments:

1.       line 51: please explain the abbreviation FDA - each abbreviation must be explained the first time it is used

2.       line 53: the term "alternative approaches" has a negative meaning - it can be understood as unconventional therapeutic techniques that are not accepted by expert groups and are not recommended for therapy - please explain precisely what therapeutic strategies you mean or remove this term

3.       line 71: "Subjects of both sexes aged 18 years or older" - please indicate the number of women and men

4.       line 92: what method was used to test total IgE (please provide the method and analyzer - due to the sensitivity and specificity of analytical methods, this is not without significance

5.       line 93: "radio allergosorbent test - RAST" - please write what specific method these tests were performed. The RAST method ("RADIO" - immunological - this is a method in which the detection of the final product is carried out by measuring the intensity of gamma radiation from a radioactive tracer) has not been performed for over 30 years, it has been successfully replaced by safer immuno-fluorescence and immuno- colorimetric. These are not radioimmunological methods (i.e. RAST) - due to the sensitivity and specificity of analytical methods, this is not without significance

6.       please indicate whether total and specific IgE tests "before" and "after" Dupilumab treatment were always performed using the same method and on the same analyzer

7.       please indicate whether specific IgE in all patients was performed in the same analytical system

8.       please indicate whether total IgE in all patients was performed in the same analytical system

9.       please specify how the material for testing was collected, how it was processed before the test, how it was stored until testing and what kind of material it was

10.   please specify for which allergens specific IgE was determined and why these allergens were selected

11.   line 94-95: "Pru p1, Pru p3, Ara h1, Ara h2, Ara h3, Ara h9, Cor a8, TriA14/TriA19" please write the allergen components correctly; e.g. not Ara h1, but Ara h 1 (there is always a space before the number)

12.   line 96: "Results were expressed in kUI/L." - please write the unit correctly - there is no such unit as kUI/L - total IgE is usually expressed as kIU/L and specific IgE in kUA/L.

13.   Table 3: the clinical value of sIgE concentration below 0.35 kUA/L cannot be assessed if it is not known what method (in what analytical system) it was measured - this is related to the analytical sensitivity and specificity of various sIgE measurement methods for in vitro diagnostics; Moreover, some of the concentrations given in the table are extremely low and probably have no clinical significance for these patients.

14.   please indicate whether the demonstrated hypersensitivities (detectable sIgE) were accompanied by clinical symptoms of allergy to these allergens and whether clinical improvement in these allergies was achieved after treatment with Dupilumab

15.   Please indicate the weaknesses and limitations of this study

16.   please format the references according to the journal's requirements

Author Response

Reviewer 3

Dear Reviewer 3,

Thank you for your careful revision of our manuscript and for your constructive comments. We have considered each issue raised and made changes to the manuscript accordingly; our responses are provided point by point below.

  1. line 51: please explain the abbreviation FDA - each abbreviation must be explained the first time it is used

Thank you, the abbreviation has been explained on first mention. Please see in the text: “…Food and Drug Administration (FDA)…”

  1. line 53: the term "alternative approaches" has a negative meaning - it can be understood as unconventional therapeutic techniques that are not accepted by expert groups and are not recommended for therapy - please explain precisely what therapeutic strategies you mean or remove this term

Thank you; to avoid potential confusion we have rephrased this term to: “combination strategies with allergen immunotherapy”

  1. line 71: "Subjects of both sexes aged 18 years or older" - please indicate the number of women and men

Thank you, we have added this information: “Subjects of both sexes aged 18 years or older (11 females and 9 males).”

  1. line 92: what method was used to test total IgE (please provide the method and analyzer - due to the sensitivity and specificity of analytical methods, this is not without significance

Thank you, we have added this information. Please see in the text “Total serum IgE using ImmunoCAP total IgE test with Phadia Laboratory System (Thermo Fisher Scientific Inc., Waltham, MA, USA).”

  1. line 93: "radio allergosorbent test - RAST" - please write what specific method these tests were performed. The RAST method ("RADIO" - immunological - this is a method in which the detection of the final product is carried out by measuring the intensity of gamma radiation from a radioactive tracer) has not been performed for over 30 years, it has been successfully replaced by safer immuno-fluorescence and immuno- colorimetric. These are not radioimmunological methods (i.e. RAST) - due to the sensitivity and specificity of analytical methods, this is not without significance

We apologize for using the outdated term RAST to refer to an allergy diagnostic assay. We have removed this term and added a description of the tests that were used: “Specific IgE to food and respiratory allergens using ImmunoCAP fluores-cence enzyme immunoassay (FEIA) system (Thermo Fisher Scientific Inc., Waltham, MA, USA);”

  1. please indicate whether total and specific IgE tests "before" and "after" Dupilumab treatment were always performed using the same method and on the same analyzer

Thank you, we confirm that the same analytical system was used at T0 and at T1. Please see below.

  1. please indicate whether specific IgE in all patients was performed in the same analytical system

Thank you, we confirm that the same system was used in all instances. Please see below.

  1. please indicate whether total IgE in all patients was performed in the same analytical system

Thank you, we confirm that the same system was used in all instances. Please see in the text: “Total and specific IgE were measured using the same method and analytical system for all patients.”

  1. please specify how the material for testing was collected, how it was processed before the test, how it was stored until testing and what kind of material it was

Thank you, we have added a relevant clarification: “Serum samples were obtained from all patients at the end of a visit before initiation of treatment (T0) and after 16 weeks of dupilumab (T1) and were sent daily to our hospital’s central diagnostic laboratory for routine evaluation.”

  1. please specify for which allergens specific IgE was determined and why these allergens were selected

Thank you we have added an explanation in the methods: “…food allergens were tested according to anamnesis and included peanut, hazelnut, almond, walnut, cereals, peach; the major respiratory allergens tested in all patients were Cupressus sem-pervirens, Dermatophagoides pteronyssinus, Dermatophagoides farina, Lolium perenne, Parietaria Judaica, Poa pratensis.”

  1. line 94-95: "Pru p1, Pru p3, Ara h1, Ara h2, Ara h3, Ara h9, Cor a8, TriA14/TriA19" please write the allergen components correctly; e.g. not Ara h1, but Ara h 1 (there is always a space before the number)

We apologize for this mistake that has been corrected throughout the manuscript.

  1. line 96: "Results were expressed in kUI/L." - please write the unit correctly - there is no such unit as kUI/L - total IgE is usually expressed as kIU/L and specific IgE in kUA/L.

Thank you, we corrected the abbreviations: “Results were expressed in kilounits per liter (kU/L) for total IgE and in kilounits of aller-gen-specific IgE per liter (kUA/L) for specific IgE, with detection down to 0.1 kUA/L.”

  1. Table 3: the clinical value of sIgE concentration below 0.35 kUA/L cannot be assessed if it is not known what method (in what analytical system) it was measured - this is related to the analytical sensitivity and specificity of various sIgE measurement methods for in vitro diagnostics; Moreover, some of the concentrations given in the table are extremely low and probably have no clinical significance for these patients.

Thank you for your comment: to allow correct interpretations of our results, we have reported in the methods which test was used and the lower value of detection. Please see in the text: “Total serum IgE using ImmunoCAP total IgE test with Phadia Laboratory System (Thermo Fisher Scientific Inc., Waltham, MA, USA). … with detection down to 0.1 kUA/L.”

  1. please indicate whether the demonstrated hypersensitivities (detectable sIgE) were accompanied by clinical symptoms of allergy to these allergens and whether clinical improvement in these allergies was achieved after treatment with Dupilumab

Thank you, we have added the following statement to the Results section: “A detailed dietary history was obtained from each patient at baseline to confirm the correlation between allergic manifestations following food intake and the positive values of the related specific IgEs. Although our patients didn’t undergo oral food challenge (OFC) while under dupilumab treatment, they were monitored to collect any adverse reaction and no clinical symptoms associated to food exposure were reported throughout the study.”

  1. Please indicate the weaknesses and limitations of this study

Thank you, limitations to our study have now been grouped and discussed in the conclusion: “However, it is presently unknown to what extent and how long allergen tolerance can persist if treatment with dupilumab is eventually interrupted. This limitation is inherent to the current chronic use of the drug and future studies are necessary to elucidate this aspect. Moreover, the role of differential administration schedules of dupilumab is still investigational and dose-spacing regimens every 4 weeks are being studied in adoles-cents with AD [28]. Additional limi-tations of our study are the small sample size due to the selectivity of the inclusion criteria and the inability to perform an OFC test, which is considered the diagnostic gold standard, even though all patients were on unrestricted diet and no adverse events were registered during the study.”

  1. please format the references according to the journal's requirements

Thank you, the reference list has been rewritten according to the Journal’s guidelines and a new reference 28 has been added.

Round 2

Reviewer 1 Report

Comments and Suggestions for Authors

The authors have extensively reviewed and improved the manuscript .

Some few small corrections needed - Line 20- allergic sensitization  

Line 52 - involved in food allergies instead of OAS ,  Line 184- suffering more severe forms or eosinophilic gastroenteritis , Line 185- delete systemic , just anaphylaxis

Author Response

Reviewer 1

The authors have extensively reviewed and improved the manuscript .

Dear Reviewer,

Thank you for your further efforts on our manuscript. Please find our responses to your comments below.

Some few small corrections needed - Line 20- allergic sensitization 

Thank you, this was changed as suggested: “on the allergic sensitization profile”

Line 52 - involved in food allergies instead of OAS , 

Thank you, we have edited according to your suggestion to “involved in food allergies”

Line 184- suffering more severe forms or eosinophilic gastroenteritis ,

Thank you, we have rewritten this sentence according to your suggestions; please see in the text: “to patients suffering more severe forms or eosinophilic gastroenteritis.”

Line 185- delete systemic , just anaphylaxis

Thank you, we changed to “at risk of anaphylaxis.”

Reviewer 3 Report

Comments and Suggestions for Authors

Thank you for the changes you made, but I have a lot of other comments:

1.       line 110: please start from the same level as line 108 - total IgE and specific IgE are completely different analytical tests; sIgE is not part of tIgE from the analytical point of view

2.       line 117: "Specific molecular determinants for component resolved diagnosis" - please write precisely that "concentration of IgE specific for these components" was measured, but not "concentration of these components" (as it is written now) - this is a significant difference

3.       line 120: how was the serum collected, in what system (open or closed), into what tubes, how long was the blood clotted, how was it centrifuged (time, temperature, acceleration), how was it stored until the assay (temperature and how long), whether the patients were fasting or not, at what time was the blood collected - please answer these questions, it is a very important part of the methodology

4.       line 125: "IgE, with detection down to 0.1 kUA/L." - please change to "IgE. The overall limit of quantitation  for allergen specific IgE antibodies is 0.1 kUA/L (analytical sensitivity). The cross-reactivity with other human immunoglobulins is non-detectable at physiological concentrations of IgA, IgD, IgM and IgG (analytical specificity)." - please indicate the source of this information (test specification according to manufacturer's document)

5.       Line 125: please complete with analytical sensitivity and specificity (according to test specifications) for total IgE. - please indicate the source of this information (test specification according to manufacturer's document)

6.       Table 2 and Table 3 - the last lines - what is hidden under "TOTAL"?

7.       Table 2 – line “Parietaria judaica 1.09 (0;100) 0.26 (0;96.8)” and Table 3: line: "Pru p 1 0.085 (0.02; 0.12) 0.025 (0; 0.07)" and line  "Ara h 1 0.08 (0.02; 0.17) 0.02 (0; 0.06)" - it is not possible to measure specific IgE concentration below 0.1 kUA/L in the ImmunCAP system - because the sensitivity threshold of this method is 0.1 kUA/L (https://dfu.phadia.com/Data/Pdf/56cb2b6389c23251d0d2b2ff.pdf) - all values ​​below 0.1 kUA/L should be reported as < 0.1 kUA/L - values ​​in the range of 0-0.1 kUA/L are not treated as quantitative (it can be either background and interference) - the presented results require re-analysis.

8.       Table 3: please write the names of all components correctly, e.g. it is "Pru p_1" and should be "Pru p 1" (without "_" before the number)

9.       Table 2 and Table 3 - please change "allergen" to allergen (sIgE) and "Molecular component" to "Molecular component (sIgE)"

10.   Dear authors, please write who exactly performed these tests and indicate the person performing the laboratory analyses and their affiliation as the next author of this manuscript or provide information that it was possible to use the results of routine analyses for publication without the knowledge and consent of the laboratory. Without the results of laboratory analyses, your manuscript would not have been created.

Author Response

Reviewer 3

Thank you for the changes you made, but I have a lot of other comments:

Dear Reviewer,

Thank you for your careful revision of our manuscript and for your constructive comments. We have carefully considered each issue raised and modified the paper accordingly. Please find our responses below.

  1. line 110: please start from the same level as line 108 - total IgE and specific IgE are completely different analytical tests; sIgE is not part of tIgE from the analytical point of view

Thank you, we apologize for this formatting mistake which has now been fixed in the text.

  1. line 117: "Specific molecular determinants for component resolved diagnosis" - please write precisely that "concentration of IgE specific for these components" was measured, but not "concentration of these components" (as it is written now) - this is a significant difference

Thank you, we have rewritten this expression to prevent a potential source of confusion; please see in the text: “Concentration of IgE specific for molecular determinants”

  1. line 120: how was the serum collected, in what system (open or closed), into what tubes, how long was the blood clotted, how was it centrifuged (time, temperature, acceleration), how was it stored until the assay (temperature and how long), whether the patients were fasting or not, at what time was the blood collected - please answer these questions, it is a very important part of the methodology

Thank you for your constructive comment; this information was added in the methods section. Please see in the text: “Peripheral venous blood in non-fasting condition was collected into closed-system BD Vacutainer SST II Advance 5.0 mL tubes (BD, Franklin Lakes, NJ, USA) and allowed a minimum of 30 minutes to clot; afterwards it was centrifuged at 3000 g for 5 minutes – with standard acceleration and soft deceleration setting on Medifuge Centrifuge (Thermo Fisher Scientific Inc., Waltham, MA, USA) – at room temperature between 20-25° C and stored in a refrigerator (2-8° C) until the assay was performed according to the routine laboratory workflow for no longer than one week.”

  1. line 125: "IgE, with detection down to 0.1 kUA/L." - please change to "IgE. The overall limit of quantitationfor allergen specific IgE antibodies is 0.1 kUA/L (analytical sensitivity). The cross-reactivity with other human immunoglobulins is non-detectable at physiological concentrations of IgA, IgD, IgM and IgG (analytical specificity)." - please indicate the source of this information (test specification according to manufacturer's document)

Thank you for your comment. This information was added, together with a citation of the relevant source. Please see: “The overall limit of quantitation for allergen specific IgE antibodies is 0.1 kUA/L (analytical sensitivity). The cross-reactivity with other human immunoglobulins is non-detectable at physiological concentrations of IgA, IgD, IgM and IgG (analytical specificity) [12].”

12.ImmunoCAP® Specific IgE Fluoroenzymeimmunoassay Directions for Use, Available online: https://dfu.phadia.com/Data/Pdf/56cb2b6389c23251d0d2b2ff.pdf (accessed on 04 August 2024).

  1. Line 125: please complete with analytical sensitivity and specificity (according to test specifications) for total IgE. - please indicate the source of this information (test specification according to manufacturer's document)

Thank you for your comment. This information was added, together with a citation of the relevant source. Please see: “The overall limit of quantitation for total IgE antibodies is 2 kU/L (analytical sensitivity). The cross-reactivity with other human immunoglobulins is non-detectable at physio-logical concentrations of IgA, IgD, IgM and IgG (analytical specificity) [11].”

11.ImmunoCAP™ Total IgE Fluoroenzymeimmunoassay Directions for Use. Available online: https://dfu.phadia.com/Data/Pdf/5be5621c89c2320848d710f6.pdf (accessed on 04 August 2024).

  1. Table 2 and Table 3 - the last lines - what is hidden under "TOTAL"?

Thank you for your comment. We have added a clarification below each table: “* The row “total” considers all specific IgEs to respiratory allergens.” and “* The row “total” considers all specific IgEs to molecular components.”

  1. Table 2 – line “Parietaria judaica 1.09 (0;100) 0.26 (0;96.8)” and Table 3: line: "Pru p 1 0.085 (0.02; 0.12) 0.025 (0; 0.07)" and line"Ara h 1 0.08 (0.02; 0.17) 0.02 (0; 0.06)" - it is not possible to measure specific IgE concentration below 0.1 kUA/L in the ImmunCAP system - because the sensitivity threshold of this method is 0.1 kUA/L (https://dfu.phadia.com/Data/Pdf/56cb2b6389c23251d0d2b2ff.pdf) - all values ​​below 0.1 kUA/L should be reported as < 0.1 kUA/L - values ​​in the range of 0-0.1 kUA/L are not treated as quantitative (it can be either background and interference) - the presented results require re-analysis.

Thank you, we apologize for this mistake: median and ranges were recalculated accordingly, as were the p values in the results. Please see table 2 and table 3 as well as the results in the text.

  1. Table 3: please write the names of all components correctly, e.g. it is "Pru p_1" and should be "Pru p 1" (without "_" before the number)

Thank you, the underline sign was due to the track changes function in Word and it has now been removed so only the space is displayed.

  1. Table 2 and Table 3 - please change "allergen" to allergen (sIgE) and "Molecular component" to "Molecular component (sIgE)"

Thank you for your suggestion, we have changed the first row of both tables accordingly.

  1. Dear authors, please write who exactly performed these tests and indicate the person performing the laboratory analyses and their affiliation as the next author of this manuscript or provide information that it was possible to use the results of routine analyses for publication without the knowledge and consent of the laboratory. Without the results of laboratory analyses, your manuscript would not have been created.

Thank you, we have added the following information to our Ethics statement mentioning the name of the Laboratory and that only routine tests were performed: “Only the results of routinely performed analyses were used for the present study, therefore addi-tional specific consent or approval from our local laboratory “HUB di laboratorio – Polo Ospedaliero – Azienda Sanitaria Locale Roma 1” was not required.”

Round 3

Reviewer 3 Report

Comments and Suggestions for Authors

I thank the authors for their cooperation. I have no further comments.